# Evaluation of the Effectiveness of Iontophoresis with Perskindol Gel in Patients with Osteoarthritis of the Knee Joints

**DOI:** 10.3390/ijerph19148489

**Published:** 2022-07-12

**Authors:** Agnieszka Dakowicz, Zofia Dzięcioł-Anikiej, Anna Hryniewicz, Małgorzata Judycka, Mariusz Ciołkiewicz, Diana Moskal-Jasińska, Anna Kuryliszyn-Moskal

**Affiliations:** 1Department of Rehabilitation, Medical University of Bialystok, M. Curie-Sklodowska Str. 24 A, 15-276 Białystok, Poland; dzieciol.zofia@gmail.com (Z.D.-A.); anna.hryniewicz@umb.edu.pl (A.H.); malgorzata.judycka@op.pl (M.J.); marjanc40@gmail.com (M.C.); akuryl@umb.edu.pl (A.K.-M.); 2Department of Clinical of Phonoaudiology and Speech Therapy, Medical University of Bialystok, 15-276 Białystok, Poland; diana.moskal-jasinska@umb.edu.pl

**Keywords:** knee osteoarthritis, iontophoresis with Perskindol gel

## Abstract

Introduction: Osteoarthritis (OA) is one of the most common causes of pain in the musculoskeletal system leading to disability. The basic principle of the therapy is the simultaneous use of pharmacological and non-pharmacological treatments. The aim of this study was to evaluate the effectiveness of galvanic and iontophoresis treatments with Perskindol Active Classic Gel (Perskindol) in patients with OA of the knee joints. Moreover, a comparative evaluation of the effectiveness of the application was performed depending on the selection of the active electrode. Material and Methods: The study included 100 patients with gonarthrosis, treated at the Rehabilitation Clinic of the Białystok University Hospital. Three groups were randomly selected: in group I (*n* = 33), anodic galvanic treatment was applied, group II (*n* = 33) received iontophoresis with Perskindol gel from the negative pole (“−” iontophoresis), and group III (*n* = 34) received iontophoresis with Perskindol gel from the positive pole (“+” iontophoresis). The VAS, the Laitinen questionnaire, the Lequesne Index, the Lysholm questionnaire, and the SF-36v2 health survey were used for the clinical evaluation of the patients. Results: In the group of patients who underwent iontophoresis with the use of Perskindol gel introduced from the positive pole, a statistically significant improvement was shown in all the assessed parameters in comparison to the patients who underwent anodic galvanic treatment. Conclusions: The most favorable effect of iontophoresis was observed in the case of iontophoresis with Perskindol gel introduced from the positive pole.

## 1. Introduction

Osteoarthritis (OA) is the most common disease of the musculoskeletal system, leading to impaired performance in daily life, especially among the elderly. According to the WHO, it is one of the most common causes of disability in developed countries. OA can affect any joint, although the knee joints suffer from it most often [1,2].

Pathological lesions in the course of OA are the result of an imbalance between the processes of repair and damage to the cartilage tissue. Although the loss (destruction) of the cartilage tissue is fundamental, the disease affects the entire joint. The disease is progressive, conditioned by individual variability and external factors. Clinical symptoms include pain, stiffness, limited mobility, and muscle weakness. Long-term consequences include impairment of physical activity and physical condition, deterioration in the quality of sleep, fatigue, depression, and disability [3,4].

OA therapy strategies include combined pharmacological and non-pharmacological treatments [5,6,7].

The aim of the therapy, in addition to pain control, is to improve joint function and activate the processes of cartilage repair. Much attention is paid to the role of the patient in establishing an individual therapy plan, including education of both the patient and his or her immediate environment, providing orthopedic equipment, and rehabilitation. Physiotherapeutic treatment should be implemented as early as possible, and in the first stage of treatment [6,7]. Physical therapy plays an important role in the treatment, and its use helps to reduce pain, and relieve increased muscle tension and swelling. In turn, this allows the range of motion in the joint to be increased, which improves general fitness [8].

The selection of methods and the course of treatment depend on a number of factors, including age, sex, the degree of advancement of degenerative changes, and coexisting diseases. The therapeutic approach should be individually tailored to the needs and physical condition of the patient, and aimed at improving the quality of life and postponing or, if possible, avoiding surgery [7,8]. Physical procedures with the use of direct current, such as galvanic treatment and iontophoresis, are widely used in the treatment of OA [8,9,10,11,12].

Iontophoresis, due to the introduction of medicinal ions of certain chemical compounds undergoing electrolytic dissociation by force of an electric field, thereby reaching deeper tissues through the skin or mucosa, is a treatment having strong analgesic, anti-inflammatory, and anti-swelling effects. Proper selection of the active electrode is crucial. If an anion is the active ingredient of the drug, the active electrode under which the drug should be placed will be the cathode (“−”), whereas if a cation is the active substance, the drug will be introduced from the anode (“+”). The ions of the substance penetrate the skin through the path of least resistance, i.e., mainly through the sweat glands. It should be emphasized that the advantage of iontophoresis is the fact that a high concentration of the drug can be obtained in the target tissue without the need for oral administration, resulting in a reduced risk of overdose and side effects [8,13,14,15].

Several factors influence the effectiveness of the iontophoresis treatment. These include, among others, the physicochemical properties of the compound, i.e., the particle size and charge, the concentration of the substance, and/or the presence of other ions in the preparation. The equipment used for the procedure, the intensity and type of the current flow, the type of electrodes used, and the duration of the procedure are also important. The effect of the therapy also depends on biological factors, such as skin surface area and temperature, and regional blood flow [8].

In the present work, Perskindol Active Classic Gel with a double cooling and warming effect was used for the iontophoresis treatment. The unique, two-step effect of Perskindol gel relies on the cooling and pain-reducing effect due to the menthol content, which activates the cold receptors, and cools and raises the sensitivity threshold of local pain receptors. The next stage is a long-lasting warming and relaxing effect impacting the tense muscles. This is possible due to the essential oils, which have a warming effect and improve microcirculation, positively influencing local metabolism and fostering tissue regeneration [16,17,18].

The natural active ingredients of Perskindol gel, such as menthol, pine needle essential oil, and gaultheria and terpinolen essential oils, act directly on the tissues by penetrating the epidermal barrier. The essential oils in the Perskindol preparation reduce the intensity of pain and have anti-inflammatory properties. In addition, massage reduces muscle tension, enhancing the effect of reduced pain and discomfort. Moreover, essential oils positively influence regeneration of periarticular tissues [17,18,19,20,21,22,23,24,25,26,27,28,29,30,31,32,33,34,35,36]. To summarize, due to its unique properties, Perskindol gel is an effective and safe alternative to topically applied non-steroidal anti-inflammatory drugs (NSAIDs).

### 1.1. Assumptions and Goals of the Study

The aim of the study was to evaluate the effectiveness of galvanic treatment and iontophoresis treatment using Perskindol gel in patients with OA of the knee joints. The intensity of pain, knee joint efficiency and functional efficiency of the lower limb, and the quality of life of the patients were assessed.

Due to the lack of data on the recommendations regarding the selection of an appropriate active electrode in the treatment of iontophoresis with Perskindol gel, the next goal was to determine the optimal current polarity for the introduction of the preparation. Thus, a comparative evaluation of the effectiveness of the preparation, depending on the choice of the active electrode, was performed.

### 1.2. Methodology

The study included 100 patients (72 women and 28 men) treated at the Rehabilitation Clinic of the University Hospital of Białystok with diagnosed osteoarthritis of the knee joints. The severity of the lesions was determined according to the Kellgren–Lawrence scale [37,38,39,40,41]. Among the respondents, more than half (56.58%) were patients with 2nd degree degenerative lesions, 27.27% had 1st degree degenerative lesions, and 16.15% had 3rd degree degenerative lesions. The mean age of the patients was 56.16 years. The right knee was affected in 27 patients, the left knee in 29 patients, and both knee joints in 44 patients. The mean duration of pain intensity was 55.78 months (about 4.5 years).

Three groups were randomly selected among the study participants. In group I (*n* = 33), anodic galvanic treatment was performed, whereas in groups II (*n* = 33) and III (*n* = 34), the iontophoresis procedure was performed with the use of Perskindol gel, with group II having the preparation administered from the negative pole (“−” iontophoresis) and group III from the positive pole (“+” iontophoresis).

Each of the three groups was included in a 10-day physical therapy treatment program that included a weekend break. The PHYSIOMED-Expert stimulator was used to carry out the therapy. The treatment time was 15 min on the first day and 20 min on each subsequent day. The dose of direct current was adjusted according to the patient’s sensation, but the limit value of the current was not exceeded (anodic galvanic treatment up to 0.2 mA/cm^2^ of the active electrode surface area, iontophoresis up to 0.1 mA/cm^2^).

The patients were informed about the course of the therapy and gave their informed consent to participate in the research. A clinical evaluation was performed before and after initiation of the treatment. The conducted research was approved by the Bioethics Committee of the Medical University of Bialystok (No. R-I-002/002/463/2019).

### 1.3. Clinical Evaluation

Each patient was interviewed and underwent a physical examination, taking into account the scales and questionnaires described below [10,12,42,43,44,45,46]:

Pain intensity assessment on the Visual Analogue Scale (VAS)—A straight line was used with the starting point of 0 indicating no pain, and the end point of 10 indicating unbearable pain.

The Laitinen questionnaire was used to assess pain intensity and frequency, the frequency of taking painkillers, and the limitation of physical activity. Each indicator was assessed according to a score from 0 to 4, where 0 meant there was no problem, and 4 indicated the maximum level of problem severity.

The Lequesne Index was used to assess pain intensity and function of the knee joint. It consists of 10 survey questions given to patients with osteoarthritis of the knee. It includes five questions about pain or discomfort, one question about the maximum distance traveled, and four questions about daily activities. The entire questionnaire is rated on a scale from zero to 24, with lower scores indicating less impairment of function.

The Lysholm questionnaire is a 100-point scale used to make a subjective assessment of the functional capacity of the knee joint. In the final evaluation, patients having a higher total score have a better functional capacity.

SF-36v2 Health Survey—This questionnaire was used for a subjective assessment of the patient’s health and quality of life. The information obtained was summed to measure mental health (Mental Component Summary—MCS) and physical health (Physical Component Summary—PCS).

### 1.4. Statistical Analysis

The following parameters were used to present numerical and ordinal variables: median, lower quartile (Q1), upper quartile (Q3), and minimum and maximum.

In the statistical analysis, due to the non-existence of normality of distributions, non-parametric methods were used. When comparing the two dependent variables, the Wilcoxon order of pairs test and the sign test were used. When comparing the three independent variables, the Kruskal–Wallis ANOVA rank test was used with the post-hoc test for multiple comparisons of the rank mean for all the trials. The chi-square test of independence was used to check the relationship between the qualitative features. The McNemar test was used to analyze the relationship between the nominal dependent variables. Differences at the level of *p* < 0.05 were accepted as statistically significant.

## 2. Results

### 2.1. The VAS Pain Assessment Scale

The pain intensity level (VAS) before treatment did not differ significantly between the three groups.

After the anodic galvanic treatment, pain intensity was reduced by an average of 2 (median = −2). After “−” iontophoresis, it was also reduced by 2 (median = −2), whereas after “+” iontophoresis, pain intensity decreased by an average of 3 (median = −3). The use of “+” iontophoresis caused a statistically significant reduction in pain intensity compared to galvanic treatment (Kruskal–Wallis post-hoc test, *p* = 0.012) (Figure 1, Table 1).

### 2.2. The Laitinen Questionnaire

Before the treatment, there were no statistically significant differences between the groups.

After the anodic galvanic treatment, a reduction in pain intensity by an average of 2 (median = −2) was shown. After “−” iontophoresis, pain intensity also reduced by 2 (median = −2), whereas after “+” iontophoresis, pain intensity decreased by an average of 3 (median = −3). A statistically significant better effect in terms of pain reduction was obtained after “+” iontophoresis compared to anodic galvanic treatment (Kruskal–Wallis post-hoc test, *p* = 0.044). A statistically significant better effect in reducing pain intensity was obtained after “+” iontophoresis compared to “−” iontophoresis (Kruskal–Wallis post-hoc test, *p* = 0.001) (Figure 1, Table 1).

### 2.3. The Lequesne Index

Before the treatment, there were no statistically significant differences between the groups.

After the anodic galvanic treatment, the score decreased by an average of 2 (median = −2). After “−” iontophoresis, the score decreased by 3 (median = −3), whereas after “+” iontophoresis, the score decreased by an average of 5 (median = −5). A statistically significant better effect was obtained with the use of “+” iontophoresis compared to anodic galvanic treatment (Kruskal–Wallis post-hoc test, *p* = 0.033) (Figure 1, Table 1).

### 2.4. The Lysholm Knee Index

Before the treatment, there were no statistically significant differences between the three groups.

After the anodic galvanic treatment, the score increased on average by 6 (median = 6). After “−” iontophoresis, the score increased by 9 (median = 9), whereas after “+” iontophoresis, the score increased by an average of 12 (median = 12). A higher score was obtained with “+” iontophoresis than with anodic plating, with the statistical significance maintained compared to galvanic treatment (Kruskal–Wallis post-hoc test, *p* = 0.014) (Figure 1, Table 1).

**The Physical Component Summary (PCS).** There were no statistically significant differences between the applied forms of treatment.**The Mental Component Summary (MCS).** Statistically, the MCS score before treatment did not significantly differ between the three groups. After the galvanic treatment, the MCS score decreased on average by −0.13 (median = −0.13). After “−” iontophoresis, the score increased by 3.78 (median = 3.78), whereas after “+” iontophoresis, it increased by an average of 5.10 (median = 5.10). A statistically significant and better effect of MCS growth was obtained after “+” iontophoresis compared to anodic galvanic treatment (Kruskal–Wallis post-hoc test, *p* = 0.007) (Figure 1, Table 1).

## 3. Discussion

According to the current recommendations regarding the conservative (non-surgical) treatment of OA, the basic principle of therapy is the simultaneous use of pharmacological and non-pharmacological treatments [6,7].

In view of the numerous limitations and side effects associated with pharmacological treatment, increasing attention is being paid to non-pharmacological methods of treatment that ensure the effectiveness and safety of the therapy [14].

In the present study, the effectiveness of direct current treatments (galvanic treatment and iontophoresis with the use of Perskindol gel) was assessed in a group of 100 patients with osteoarthritis of the knee joints. The following parameters were assessed: pain intensity, functional efficiency, and quality of life. Moreover, an attempt was made to determine the most favorable current polarity in the iontophoresis procedure with the use of Perskindol gel.

In the first group, anodic galvanic treatment was performed. In the second group, iontophoresis was performed with the use of Perskindol gel, with the preparation administered from the negative pole of the current (“−” iontophoresis), whereas in the third group, the iontophoresis procedure was performed with the same agent given from the positive pole (“+” iontophoresis).

The literature shows that applying direct current leads to an improvement in vascular flow and a decrease in neuromuscular excitability. Additionally, iontophoresis introduces healing ions into the tissues. The great advantage is the non-invasive nature of the treatment and the fact that even a small concentration of the preparation can bring about the desired effect. It is not without significance that the medicinal product can be administered directly onto the affected tissue, bypassing the internal organs and avoiding side effects [8,13,15].

Moreover, the administration of drugs with the use of iontophoresis is recognized as an alternative in the treatment of OA [14].

To our knowledge, no studies have been performed with Perskindol gel administered via the skin using iontophoresis. The subject of the present study was the evaluation of the clinical effectiveness of iontophoresis in patients with OA of the knee joints.

The literature quotes studies on the use of iontophoresis with NSAIDs and steroid drugs in the treatment of patients with OA [8,9,10,11,12,13,42]. A previous study focused on the determination of the current polarization suitable for the introduction of ketoprofen into the rat knee joint [47]. Existing studies of NSAIDs showed improvements in walking speed, pain, and active range of motion after 5% Ibuprofen iontophoresis in the management of KO [10], and a significant decrease in pain and a significant increase in functional ability and the strength of knee extensor muscles after iontophoresis with Piroxicam gel [11]. Moreover, an evaluation of iontophoretic delivery of a cationic ketoprofen choline chloride showed significant pain relief and a reduction in knee inflammation in an osteoarthritic rat model [47].

Perskindol gel, due to the topical application of essential oils such as levomenthol, wintergreen oil, or Scots pine oil, has a multidirectional impact on tissues [17,18,19,20,21,22,23,24,25,26,27,28,29,30,31,32,33,34,35,36].

In the case of patients with OA, the analgesic effect is extremely important, as it reduces the sensitivity of nociceptor receptors and increases the pain threshold, in addition to stimulating and increasing the activity of thermoreceptors and sensory fibers in the skin and subcutaneous tissue [17,18,22,30,33,34,35].

Although OA has the limited characteristics of an inflammatory disease of a local nature, the anti-inflammatory effects of Perskindol also play an important role. In addition to direct inhibition of the inflammatory process (salicylates in goltery oil), inhibition of the release of inflammatory mediators and the impact on the synthesis of prostaglandins play an important role in treatment. Furthermore, levomenthol, one of the ingredients of Perskindol, increases the absorption of salicylates, which contributes to the enhancement of the anti-inflammatory effect [19,34,36].

The significant therapeutic importance of levomenthol is clear in the activation of endogenous opioid-dependent analgesic pathways which, given their local anesthetic properties, allow an analgesic effect to be obtained. This helps to increase the range of joint mobility. Moreover, increasing skin blood flow improves blood supply and local metabolism in the treated tissues [17,18,30].

Other therapeutic effects of Perskindol ingredients include the improvement in peripheral circulation and microcirculation, leading to an acceleration in metabolism in the tissues, the removal of harmful metabolic products, and an improvement in tissue oxygenation [34].

Several other effects of Perskindol ingredients should also be emphasized, such as a reduction in muscle tension, the acceleration of regenerative processes, and relaxing and stress-relieving effects through the influence on the limbic system [33].

The richest source of plant-based salicylates in Perskindol is wintergreen oil, with its analgesic and anti-inflammatory properties resulting from the inhibition of prostaglandin and nitric oxide synthesis (antioxidant effect) [19,31,32].

Pine oil and other oils such as bergamot oil, lavender oil, citrus oil, orange peel or rosemary oil also have anti-inflammatory and analgesic properties [24,25,26,27,28,29,35].

The study showed that the applied physical therapy resulted in a reduction in pain assessed on the VAS. Improvement was shown in all groups, although a better effect was obtained after “+” iontophoresis.

The Laitinen questionnaire showed a reduction in pain intensity and frequency, less frequent use of painkillers, and an improvement in physical activity in all groups of patients, although a more favorable effect was obtained after “+” iontophoresis.

In this study, the Lequesne Index was used to assess pain intensity and function of the knee joint. A higher score on this scale indicates a higher degree of impairment of the knee joint. After the therapy, the functional state was improved and pain intensity was reduced in all groups of patients. A statistically significant better effect was obtained after “+” iontophoresis.

The Lysholm questionnaire, which was also used in the study, showed an improvement in the subjective assessment of the functional capacity of the knee joints among all the respondents. In all groups, there was a statistically significant difference in scores before and after the treatment. When comparing the three treatments, a statistically significant increase in the number of points was obtained after iontophoresis from the “+” anode.

The SF-36v2 Health Survey, consisting of 36 questions, was used to assess the quality of life of patients with chronic OA of the knee joints. It provides an eight-step profile of results, allowing for a summary of the physical and mental components.

Although there were no significant differences in the sum of the physical components (PCS) between the proposed treatments, the highest median score was obtained in group III (“+” iontophoresis), of 3.91.

In terms of the mental components (MCS) score, a statistically significant increase in the score was obtained after “+” iontophoresis.

The obtained effects indicate the effectiveness of Perskindol in the treatment of patients suffering from chronic OA of the knee joints, with the best therapeutic effect obtained after iontophoresis with Perskindol gel introduced from the positive pole.

It is very important to undertake comprehensive therapy of the knee joint to prevent functional changes in other adjacent joints, e.g., in the joints of the feet, which may be affected by RA, as they play a very important role in maintaining proper body posture and movement [48,49,50].

## 4. Conclusions

In the group of patients who underwent iontophoresis with Perskindol gel introduced from the positive pole, statistically significant improvements in the degree of pain intensity, functional capacity, and performance of the knee joints, in addition to an improvement in the quality of life, was demonstrated in relation to the patients who underwent the anodic galvanic treatment.In the group of patients receiving iontophoresis with Perskindol gel introduced from the positive pole, a significant improvement was achieved in the degree of pain intensity, as assessed according to the Laitinen questionnaire, in relation to the patients treated with iontophoresis with the use of this preparation applied from the negative pole.In the studied group of patients with OA of the knee joints, the most favorable therapeutic effect was obtained using iontophoresis with the use of Perskindol gel introduced from the positive pole.

Taking into account the above results, it is worth considering the use of Perskindol gel during iontophoresis procedures in patients with osteoarthritis of the knee joint.

## Figures and Tables

**Figure 1 ijerph-19-08489-f001:**
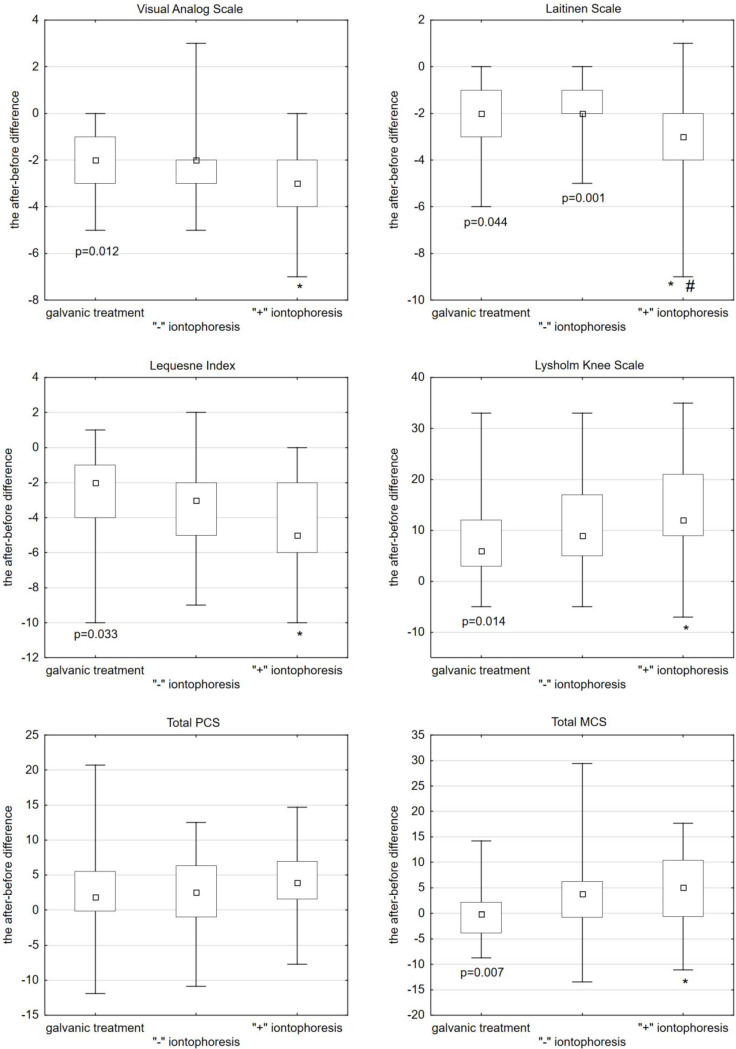
Scales and questionnaires used in the studied group of patients (differences before and after therapy). * 1 group vs. 3 group; # 2 group vs. 3 group.

**Table 1 ijerph-19-08489-t001:** Differences (before the treatment—after the treatment) in the results on the scales used to evaluate the patients. The results are presented as: median (lower quartile, upper quartile). The last column shows the results of the Kruskal–Wallis post-hoc test.

Scale	(1) Galvanic Treatment	(2) ‘’−” Iontophoresis	(3) ‘’+” Iontophoresis	1 vs. 2 1 vs. 3 2 vs. 3
VAS	−2.00 (−3.00; −1.00)	−2.00 (−3.00; −2.00)	−3.00 (−4.00; −2.00)	n/a *p* = 0.012 n/a
Laitinen Questionnaire	−2.00 (−3.00; −1.00)	−2.00 (−2.00; −1.00)	−3.00 (−4.00; −2.00)	n/a *p* = 0.044 *p* = 0.001
Lequesne Index	−2.00 (−4.00; −1.00)	−3.00 (−5.00; −2.00)	−5.00 (−6.00; −2.00)	n/a *p* = 0.033 n/a
Lysholm Knee Index	6.00 (3.00; 12.00)	9.00 (5.00; 17.00)	12.00 (9.00; 21.00)	n/a *p* = 0.014 n/a
MCS Index	−0.13 (−3.84; 2.14)	3.78 (−0.73; 6.21)	5.10 (−0.60; 10.44)	n/a *p* = 0.007 n/a

## Data Availability

The data presented in this study are available on request from the corresponding author (agadak@interia.pl).

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
