# Peer review of "Evaluation of the Effectiveness of Iontophoresis with Perskindol Gel in Patients with Osteoarthritis of the Knee Joints"

_ijerph, 2022, doi:10.3390/ijerph19148489_

Round 1
Reviewer 1 Report
This is an interesting paper describing the delivery of Perskindol gel using iontophorisis in a cohort of Participants with knee osteoarthritis. This is a first for this product and is based on evidence from the literature on the delivery of anti-inflammatory compounds using this method. The product has dual effects on reduction of pain by reducing inflammation and desensitisation of the local pain receptors through cooling ingredients and reducing muscle tension through warming ingredients. The authors have demonstrated the optimal delivery system and through a range of different instruments that there is an improvement in physical symptoms and related to this an improvement in the mental health of the participants.
Methodology: The authors mention Kellgren Lawrence grading for lesions. Could the authors please elaborate a little on what they mean by this and do they use a definition of osteoarthritis such as KL grade greater or equal to 2 (similar to that described by the Americal College of Rheumatology)?
Was there a power calculation for ensuring that this was the correct study size?
Was there any intention to stratify the study population by OA severity?
Clinical evaluation: The authors mention physical examination of the participants but there no details of what this entailed. Please expand upon this.
The Lequesne Index does not mention how many questions were in this instrument and the scale used for measurement. Please amend this information.
Results: There is no information provided on cohort demographics.
Lequesne Index - related to the previous comment it is difficult to determine from the results presented the magnitude of the change. Please review and amend.
MCS: I am unclear about what the authors mean in the final sentence of this paragraph. I think they mean a statistically significant improvement in mental health was demonstrated - please review and clarify.
In the fifth paragraph of the results, I think that there needs to be some references added to support the evidence, or if they are at the end of the paragraph, moved to support the appropriate statements.
Did the authors report any adverse events in study?
Was the physcial examination repeated?
I think paragraphs 6-8 to should merged with the information about existing studies with NSAIDs delivered by this method coming before the novel Perkindsol study, I believe that this will improve the impact of the statement.
Please provide some references for the mechanism of action of levomenthol.
Reviewer 2 Report
Dear Authors, This is an interesting study and I enjoyed reading your manuscript. There are few minor comments, which I believe can strengthen the quality of your work:
1. I see that KL scale was used to grade OA. Over 50% patients have Grade II OA. Can you elaborate/add on the KL scales for rest of the population. Also, add what is the KL for males vs females. Adding this information will provide more comprehensive understanding of the patient population included in the study.
2. Figure 1: Can you add an asterix etc. to clearly show which group is significant compared to what (similar to table) for better understanding and update legend accordingly.
3. In the discussion, please add references where function of individual components of the Gel is discussed.
4. Most of the outcome measures used in discussion are not really discussed and looks more like re-emphasizing the results section. It would be great to actually discuss these and also compare the results obtained with studies published related to Perskindol gel (if any) and with studies related to NSAIDs and corticosteroids utilizing ionotophoresis (also mention any polarity information described in these studies, if provided).
5. Add a paragraph for future studies indicating that more studies including RCTs are warranted to establish efficacy of this treatment to justify its clinical use and establish it as a replacement therapy for current pharmacological modalities and/or as first line non-pharmacological modality that should be tried.
6. Add a conflict of interest statement as a commercial product is used in this study.
